# Cryopreservation of Gametes and Embryos and Their Molecular Changes

**DOI:** 10.3390/ijms221910864

**Published:** 2021-10-08

**Authors:** Enrique Estudillo, Adriana Jiménez, Pablo Edson Bustamante-Nieves, Carmen Palacios-Reyes, Iván Velasco, Adolfo López-Ornelas

**Affiliations:** 1Laboratorio de Reprogramación Celular, Instituto Nacional de Neurología y Neurocirugía Manuel Velasco Suárez, Mexico City 14269, Mexico; jestudillo@innn.edu.mx (E.E.); ed_bustamante@ciencias.unam.mx (P.E.B.-N.); ivelasco@ifc.unam.mx (I.V.); 2Departamento de Fisiología, Facultad de Medicina, Universidad Nacional Autónoma de México, Mexico City 04510, Mexico; adijh2@gmail.com; 3Facultad de Ciencias, Universidad Nacional Autónoma de México, Mexico City 04510, Mexico; 4División de Investigación, Hospital Juárez de México, Mexico City 07760, Mexico; cyapalacios@gmail.com; 5Instituto de Fisiología Celular—Neurociencias, Universidad Nacional Autónoma de México, Mexico City 04510, Mexico; 6Hospital Nacional Homeopático, Hospitales Federales de Referencia, Mexico City 06800, Mexico

**Keywords:** egg cryopreservation, vitrification, oocytes, epigenetic changes

## Abstract

The process of freezing cells or tissues and depositing them in liquid nitrogen at –196 °C is called cryopreservation. Sub-zero temperature is not a physiological condition for cells and water ice crystals represent the main problem since they induce cell death, principally in large cells like oocytes, which have a meiotic spindle that degenerates during this process. Significantly, cryopreservation represents an option for fertility preservation in patients who develop gonadal failure for any condition and those who want to freeze their germ cells for later use. The possibility of freezing sperm, oocytes, and embryos has been available for a long time, and in 1983 the first birth with thawed oocytes was achieved. From the mid-2000s forward, the use of egg vitrification through intracytoplasmic sperm injection has improved pregnancy rates. Births using assisted reproductive technologies (ART) have some adverse conditions and events. These risks could be associated with ART procedures or related to infertility. Cryopreservation generates changes in the epigenome of gametes and embryos, given that ART occurs when the epigenome is most vulnerable. Furthermore, cryoprotective agents induce alterations in the integrity of germ cells and embryos. Notably, cryopreservation extensively affects cell viability, generates proteomic profile changes, compromises crucial cellular functions, and alters sperm motility. This technique has been widely employed since the 1980s and there is a lack of knowledge about molecular changes. The emerging view is that molecular changes are associated with cryopreservation, affecting metabolism, cytoarchitecture, calcium homeostasis, epigenetic state, and cell survival, which compromise the fertilization in ART.

## 1. Introduction

In 1776, Lazaro Spallanzani cryopreserved the first gametes, specifically spermatozoa, which regained their motility [1,2]. The cryoprotective properties of glycerol in the same cells were discovered in 1949 by Christopher Polge and Audrey Smith. Subsequently, Polge and Tim Rowson performed the first artificial insemination in cattle in 1952 [3]. The first successful cryopreservation of human spermatozoa that achieved pregnancies and live births was reported by Sherman and Bunge in 1953; interestedly, they used liquid nitrogen for cold storage [4]. Ten years later, Sherman reported no loss of motility after one year of storage [5]. In 1964, Perloff reported the storage of spermatozoa for more than five months without loss of viability [6]. Afterward, frozen–thawed spermatozoa achieved pregnancies after in vitro fertilization (IVF) in 1984 [7], in intrauterine insemination (IUI) in 1990 [8], subzonal insemination in 1992 [9], and intracytoplasmic sperm injection (ICSI) in 1994 [10].

Embryo cryopreservation precedes oocyte cryopreservation. In 1971, Whittingham reported the first cryopreservation in mouse embryos [11] and later, other species were cryopreserved [3], including cattle [12], rabbits [13], rats [14], horses [15], and non-human primates [16,17]. The first embryo cryopreservation in humans was achieved in 1983 [18] and the firsts live births were reported in 1984 [19]. All these accomplishments employed the slow-freezing method. In 1985, Rall reported cryopreservation by vitrification of mouse embryos [20] and in 1995 day-2 human embryos were vitrified, with in vitro survival [21].

Cryopreservation of human oocytes was achieved in 1988 [22] after several previous attempts. In 1992, Arav developed the “minimum droplet size” technique and earned an increase in the cooling rate, therefore avoiding the formation of ice crystals, with a reduced concentration of volume and cryoprotectants (CPA) [23]. The first birth from frozen oocytes using ICSI was reported in 1997 [24]. In 1989, vitrification of mature mouse oocytes was reported, which resulted in the birth of live animals [25]; the first human pregnancy after vitrification of oocytes was in 1999 [26]. The first ICSI was performed, trying to overcome the hardening of the zona pellucida and allow fertilization. However, the use of slow-frozen eggs yielded low pregnancy rates. For this reason, the vitrification method for egg cryopreservation became more widely used [27]. From the mid-2000s onwards, egg vitrification has improved clinical pregnancy rates when compared with slow-freezing method [28,29].

Different cryopreservation strategies have been proposed to improve cell survival and preserve cell functions [30]. Nowadays, cryopreservation can be used in various fields ranging from preserving animal biodiversity to the conservation of human tissues for multiple purposes [31,32]. Medically-assisted reproductive technologies (ART) have allowed the birth of approximately 7 million children [33,34], where cryopreservation of gametes or embryos has reduced the damage caused by hyperstimulation syndrome. The most used types of ART are conventional IVF and ICSI, which are accompanied by controlled ovarian hyperstimulation, oocyte retrieval, embryo culture, and embryo transfer [35]. Cryopreservation is a routine method, despite a relationship between ART and some disorders such as congenital anomalies, low birth weight, growth, metabolic disorders, and psychomotor or mental development delays [36]. Notably, there is an increase in rare diseases related to genomic imprinting, such as Angelman, Beckwith–Widemann, and Silver–Russell syndromes [37,38]. Thus, ART could alter the epigenetic profile of gametes and preimplantation embryos, causing alterations that persist after birth [39]. Epidemiological data show that ART-conceived children present some differences in blood pressure, body composition, and glucose homeostasis; mice present similar effects, so they can be used as models to assess ART-related alterations [39].

This review aims to address known molecular changes in cryopreservation of gametes and embryos, emphasizing human cells if the information is available. Knowledge of these changes could improve the relatively low success rate and the alterations that the resulted progeny may suffer.

## 2. Vitrification and Slow-Freezing Procedures

These techniques are habitually used in fertilization laboratories with substantial variations. These differences represent a particular drawback for determining the most appropriate method for cryopreservation [40]. However, several studies have elucidated the efficacy of each cryopreservation strategy.

### 2.1. General Principles

Gametes and embryos have unique characteristics concerning somatic cells, such as limited number, generational impact, and differences in cryosensitivity among species [41].

Cryopreservation preserves structurally intact living cells and tissue despite freezing lethality. The cooling effects are produced by the freezing of water, mainly by the concentration of the solutes in the liquid phase [42]. There are two mechanisms regarding freezing damage, the first is the formation of ice crystals, which pierce and destroy the cell, and the other is due to effects on changes in the composition of the liquid phase [43]. Extracellular ice is generally harmless, except in densely packed cells, given the impairment caused by mechanical stress that some channels can suffer [44].

In general, CPA increase the total concentration of solute and reduce the amount of ice formed [45]. Both mechanisms are crucial and their effects depend on the cell type, cooling, and thawing rate. CPA contain calcium and limit fertilization by hardening the cell membrane [46].

In slow-freezing protocol, cells rapidly efflux intracellular water and eliminate supercooling, thus preventing intracellular ice formation [43]. Water is substituted in the cytoplasm with CPA and reduces cell damage by adjusting the cooling rate with the permeability of the cell membrane [47]. This protocol employs a cooling rate of 1 °C/min with 1.0M of CPA.

The vitrification method is an alternative to the slow-freezing protocol and also prevents ice formation. There are two vitrification methods: equilibrium and nonequilibrium. The first requires the multimolar formulation of CPA and its injection into cell suspensions. The second uses a high freezing rate and lower concentrations of CPA [48]. One of the main properties of CPA viscosity is that it behaves as a solid without any crystallization. However, CPA toxicity is its most significant disadvantage. High concentrations of CPA are necessary to prevent intracellular ice crystals during vitrification. It is yet unknown whether the inhibition of crystal formation depends on CPA penetration into the cell or on the osmotic removal of water [49]. Regardless of the mechanism through which these methods avoid crystal formation, high concentrations of CPA can cause osmotic injury. Vitrification requires cooling to deep cryogenic temperatures after exposure to high concentrations of CPA with subsequent rapid cooling to avoid the ice nucleation. It is affected by sample volume, the viscosity of the sample, and cooling and warming rates [50].

In both methods, CPA must enter the entire biological system, but there are membranes for the diffusion of solutes that result in transient changes in the volumes of the cellular compartments and these are harmful. Thus, osmosis and diffusion affect the introduction and removal of CPA and the freeze-thaw process [44].

### 2.2. Effects of Cryopreservation in Germ Cells and Embryos

Previous studies suggest that embryo cryopreservation through slow-freezing decreases embryo metabolism when compared with vitrification. Human embryo survival rates and the number of intact blastomeres were higher when embryos were cryopreserved by vitrification [51]. This information highlights some advantages of vitrification over the slow-freezing strategy. In line with this evidence, a clinical study showed that vitrification produced fewer harmful effects than slow-freezing since frozen–thawed procedures performed in human embryos and submitted to slow-freezing procedures showed impaired morphology, survival, and pregnancy rates when compared with vitrification [52].

On the other hand, a limited number of reports suggest that oocytes cryopreserved by the slow-freezing method preserve a higher rate of fertilization and blastocyst formation than those submitted to vitrification [28,53]. However, several reports argue against these results [54,55,56], so more studies should be conducted to elucidate the beneficial effects of the slow-freezing strategy with more accuracy.

Although vitrification presents several advantages over slow-freezing procedures, some applications different from cryopreservation of embryos and germ cells might respond better to slow-freezing techniques. Cryopreservation by slow-freezing of ovarian tissue showed better efficacy than vitrification methods, which triggers the expression of genes associated with apoptotic pathways in human follicle oocytes. Additionally, the morphology of vitrified oocytes is worse when compared with oocytes preserved by slow-freezing [57]. Thus, slow freezing of human ovarian tissue could provide more beneficial effects than vitrification.

Cryopreservation of embryos and germ cells has provided the general notion that there is an increase in the efficacy of pregnancy and birth rates [58]. However, recent evidence challenges this idea as a retrospective study suggested that cryopreserved embryos displayed fewer live birth and pregnancy rates than the transfer of fresh embryos [59]. This information indicates that unknown molecular and cellular impairments might underlie the manifestation of these alterations that affect the integrity of embryos and germ cells. Nevertheless, although cryopreservation methods have been widely used for decades, there is a lack of information regarding their impact on the physiology and (epi)genomics of embryos and germ cells [40]. Despite the need for more knowledge about the effect of cryopreservation at molecular levels, some significant findings have been made in this field (Figure 1).

## 3. Molecular Alterations in Cryopreserved Embryos

### 3.1. Cellular Changes in Embryos

Slow-freezing and vitrification alter mitochondrial distribution in mouse embryos. Interestingly, reactive oxygen species (ROS) are higher at morula stages with the vitrification protocol, which suggests that, in addition to the cryopreservation method, the embryonic stage also determines changes in cells induced by cryopreservation [60]. Antioxidant treatments have emerged as a promising strategy to overcome cellular impairments caused by cryopreservation [61]. Resveratrol treatment of bovine blastocysts before their cryopreservation through slow freezing increases the expression of SIRT1, a protein involved in mitochondrial biogenesis. This increase is followed by an increase in the mitochondrial DNA, suggesting mitochondrial biogenesis. Finally, this treatment improved the number of hatched embryos and conception rates [62]. Further results suggest that resveratrol treatment could be administered to oocytes, and this strategy increases embryo survival [63]. These findings reveal a beneficial effect of resveratrol mediated by mitochondria that could be translated to cryopreservation of human blastocysts.

Although it is undoubtedly known that embryo viability decreases after cryopreservation, the information regarding how this procedure promotes cell death in human embryos is scarce. Recent evidence in animal studies suggests the vitrification method generates impairment in the mitochondrial function in parthenogenetic pig blastocysts. These damages, such as a decrease in mitochondrial membrane potential and overactivation of caspase genes, trigger apoptotic pathways that induce impairments in the developmental potential of embryos [64]. Cryopreserved human embryos may be affected through similar mechanisms.

### 3.2. Proteomic Changes

The vitrification technique has been widely used in recent years; however, there is a lack of information about its effects on the nervous system. Vitrification induces deregulation of the proteomic profile in the rodent brain. These proteins are involved in apoptosis, phagosome synthesis, as well as cysteine and methionine metabolism. Thus, vitrification could cause long-term impairments related to behavior or neurodegenerative diseases [65]. Interestingly, initial studies about the impact of cryopreservation on mice offspring determined that the slow-freezing of mouse embryos at two cell stages triggers an abnormal glucose and lipid metabolism. These changes are characterized by insulin resistance and alterations of the function of proteins related to glucose metabolism. Therefore, these changes could be a risk factor for metabolic diseases as type 2 diabetes [66,67,68].

### 3.3. Epigenetic Changes in Embryos

Pioneering studies have demonstrated that vitrification of murine blastocysts alters either the expression of pluripotency markers or epigenetic marks, such as histone acetylation and methylation, suggesting that human blastocysts could also be susceptible to these impairments [69].

Significantly, recent information provided further evidence regarding the effect of cryopreservation on rodent blastocyst epigenetics, as vitrification reduced epigenetic marks such as acetylation of histones H3K9 and H3K27. Remarkably, antioxidants ameliorated such histone acetylation impairments, although their administration was insufficient to re-establish the epigenetic marks to normal levels. On the other hand, antioxidants increased the survival rate of embryos and their outgrowth area and perimeter after being submitted to vitrification [70]. Despite these recent advances, it is also required to perform longitudinal studies to assess the impact of epigenetic marks caused by cryopreservation. Furthermore, information concerning the effects of slow-freezing on epigenetics is lacking, and future studies should address this issue.

Besides alterations on histone epigenetic marks, chromatin modifications by cryopreservation have also been studied to acknowledge the changes produced by this procedure in germ cells and embryos. Efforts to comprehend the impact of cryopreservation on chromatin integrity demonstrated that mouse blastocysts are more resilient to the impairments induced by cryopreservation than morulae. Interestingly, slow-freezing procedures affect chromatin integrity more than vitrification; conversely, ROS levels are higher in the vitrification protocol at morula stages than in slow-freezing. Therefore, the cryopreservation strategy differentially affects embryo integrity [60].

### 3.4. Transcriptomic and Genomic Changes in Embryos

Preclinical studies have provided some insights into the transcriptomic variations caused by these procedures in embryos. Vitrified porcine blastocysts display impaired gene expression related to glucose transport, antioxidant activity, growth, cell proliferation, or regulation of ATPase activity. Genes such as *TP53INP*, *MGMT*, and *DKK3* were upregulated in vitrified embryos, which are involved in the modulation of cell cycle or cellular defense against mutagenesis and toxicity agents [71].

Besides transcriptomic alterations, there is data concerning DNA damage in human embryos submitted to cryopreservation since vitrification and slow-freezing induce alterations of DNA integrity related to apoptotic pathways [72]. Interestingly, vitrification of human blastocysts produces fewer apoptotic cells derived from DNA damage than slow freezing procedures [73], which suggests that DNA integrity is sensitive to different types of cryopreservation protocols.

### 3.5. Offspring Changes in Embryos

Little is known about the offspring of individuals conceived by assisted reproduction, where embryos or germ cells were submitted to a cryopreservation protocol. This information is difficult to obtain since human cohorts present confounding factors that are hard to exclude [59]. Therefore, animal models could provide some insights into long-term impairments of cryopreservation. Recent studies regarding the impact of cryopreservation techniques on growth and development found no memory impairments or glial alterations in mice derived from vitrified embryos. However, there is an increase in body weight of the offspring, and this increase could lead to neuronal impairments since obesity-related genes are upregulated in the cortex [74].

## 4. Molecular Alterations in Cryopreserved Oocytes

The first birth originated from a frozen egg occurred in the 1980s, when the main purpose of performing slow-freezing was to prevent the formation of ice crystals by dehydration and gradual cooling of eggs, which was associated with the breakdown of the meiotic spindle and a loss of bipolarity and chromosome alignment [75,76]. Afterward, the egg vitrification protocol was employed. It turns liquid into a solid-state without forming ice crystals [77] and makes the cell or tissue turn into a non-crystalline glassy phase [47]. This procedure was developed to circumvent restrictive legislation related to the preservation of supernumerary embryos [78] and provides the opportunity for young women with medical conditions, such as cancer, ovarian failure, autoimmune diseases, hematopoietic stem cell transplantation, or any infertile disease, to preserve some of their eggs for later use [79]. On the other hand, egg cryopreservation can also be used for age-related elective fertility.

Cryopreservation of oocytes induces deep changes in cell physiology and transcriptomics; many studies have focused on the effect of vitrification and slow-freezing over the molecular and cellular biology of oocytes.

### 4.1. Cellular Changes in Oocytes

CPA impact on the integrity and physiology of oocytes. Particularly, the use of the cryoprotectant agent 1,2-propanediol in slow-freezing procedures induces changes in the protein expression profile of oocytes when compared with control and vitrified mouse oocytes. Furthermore, the exposure of rodent oocytes to this reagent increases intracellular calcium, ROS, zona pellucida hardening, and decreases oocyte survival [61,80].

In addition to slow-freezing, vitrification also alters the biology of oocytes as this method impairs the mouse oocyte endoplasmic reticulum organization, which can be appreciated by the lack of cortical clusters that generally appear after oocyte maturation [81]. Further information supported these results since both vitrification and slow-freezing procedures cause a decrease in cortical granules, increased vacuolization, and changes in mitochondrial morphology of human oocytes. However, zona pellucida remains morphologically intact after slow-freezing cryopreservation. Interestingly, slow-freezing can induce a higher and more sustained intracellular elevation of Ca^2+^ when compared with control and vitrified human oocytes, thus suggesting an impairment on the Ca^2+^ homeostasis of oocytes submitted to slow-freezing cryopreservation [82,83].

Noteworthy, vitrification procedures allow oocytes to restore some cellular impairments after being defrosted. Vitrification of murine oocytes with ethylene glycol (EG) and Dimethyl sulfoxide (DMSO) alters the distribution of the IP3 receptor 1 (IP3R1) at the moment of being thawed. So far, 90 min after thawing, vitrified oocytes recover the IP3R1 distribution pattern to similar levels of fresh oocytes. On the other hand, vitrified oocytes with EG and DMSO do not display altered intracellular Ca^2+^ oscillatory activity when treated with IP3 or when the oocyte fusion with sperm cells triggers this process, thus suggesting no impairment on Ca^2+^ homeostasis [84]. Further studies showed consistent results regarding the resilience of vitrified oocytes, as vitrification of mouse oocytes during their pre-antral stage displays impairment of the mitochondrial inner membrane potential, which is no longer present 12 days later after oocyte thawing [83,85].

Furthermore, EG and DMSO combination used to cryopreserve oocytes does not alter the cell number of blastocysts [84]. These results support the resistance of oocytes to CPA and highlight the relevance of designing an adequate combination of cryoprotectant agents to preserve the oocyte integrity. Finally, although some processes remain unaffected by cryopreservation, there could be others that might indeed be impaired, such as changes in the structure of the oocyte plasma membrane and the amount of intracellular lipids [86].

Another phenomenon that has been almost entirely unaddressed is the mechanisms that underlie cell death of cryopreserved human oocytes. As with embryos, there is a lack of information regarding the pathways contributing to the decreased viability of cryopreserved human oocytes [87]. Nonetheless, animal studies have provided information regarding the mechanisms of oocyte cell death after cryopreservation as bovine cumulus–oocyte complexes (COCs) that undergo apoptosis following slow-freezing and vitrification procedures [88]. Interestingly, contrasting information suggests that vitrification of COCs derived from sheep does not induce apoptosis. However, this strategy increases chromosomal abnormalities after the maturation of oocytes previously submitted to this type of cryopreservation [89]. These studies indicate that more research is required to elucidate the role of apoptosis on oocyte viability. Paradoxically, recent findings showed that genes associated with necroptosis have a role in preserving the oocyte integrity after the vitrification of mouse oocytes at metaphase 2 stage since inhibition of this pathway through the use of Necrostatin-1 decreases the survival rate of vitrified oocytes [86].

### 4.2. Proteomic Changes in Oocytes

Information related to the protein profile of human oocytes submitted to cryopreservation is needed. Most of the studies are focused on animal models. Remarkably, no differences were found in the protein profile of vitrified mouse oocytes when compared with fresh oocytes [85]. On the other hand, after cryopreservation of porcine oocytes, proteomic analyses found a total of 59 and 94 proteins that were downregulated and upregulated, respectively. The differentially expressed proteins were related to several biological processes such as metabolic processes, coagulation, heterochromatin organization, or immune responses [90]. These results highlight the importance of performing similar experiments in human oocytes to a better understanding of the biological processes mediated by proteins that could compromise the oocyte integrity after cryopreservation

### 4.3. Epigenetic Changes in Oocytes

The epigenetic changes of human oocyte vitrification remain largely unknown. However, some insights have been gained from animal experiments. Studies regarding the impact of vitrification on mouse oocytes demonstrated that enzymes related to methylation processes could be differentially affected by this cryopreservation protocol as it decreases the gene expression of DNA methyltransferase 1o (*Dnmt1o*). At the same time, histone acetyltransferase 1 (Hat1) and deacetylase 1 (Hdac1) remain with no change in their expression levels after vitrification. Interestingly, none of these enzymes display alterations in their methylation patterns, suggesting other mechanisms contributing to the downregulation of *Dnmt1o* expression [91]. Bovine blastocysts derived from vitrified oocytes show decreased levels of genome methylation. Specifically, methylation levels of promoter, exon, and intron regions were reduced when compared with controls. These results suggest differential outputs of embryo development depending on whether they were cryopreserved or not [92]. Changes in histone modifications reveal further epigenetic modifications induced by cryopreservation methods. Mouse oocytes submitted to vitrification display increased levels of histone 3 dimethylation in lysine 9 (H3K9me2); meanwhile, this process does not change acetylation levels in lysine 14 (H3K14ac). In contrast, there is also an increase in acetylation of histone 3 at lysine 5 (H3K5ac) [93]. Porcine oocytes submitted to vitrification display hyperacetylation of histone 4 two hours after thawing. Additionally, they show an increase in the methylation of histone 3 [94]. On the other hand, slow-freezing increases the tri-methylation of histone 3 at lysine 27 (H3K27me3) and decreases the trimethylation of histone H3 at lysine 4 (H3K4me3), which could impair embryo development [95]. These experiments reveal a complex alteration of epigenetic marks of the DNA and histones that could change embryo development and show a correlation between cryopreservation and epigenetic alterations. Notwithstanding, it was found that vitrification of human oocytes does not impair embryo development and does not generate methylation and hydroxymethylation at genomic scale [96]. Thus, further studies should be performed to identify epigenetic changes in human oocytes and determine whether they are partially responsible for alterations during embryo development.

### 4.4. Transcriptomic and Genomic Changes in Oocytes

Genetic material of oocytes is also sensitive to cryoprotective procedures. The cryopreservation of bovine oocytes impairs the chromosomal arrangement in metaphase II and impairs the microtubule distribution along the cytoskeleton. These issues are attenuated by the use of CPA as EG [97]. Further studies demonstrated that the Cryotop vitrification method impairs the DNA integrity in cat oocytes due to the overactivation of Caspase proteins. This activation induces apoptotic pathways, which are related to poor oocyte development. Interestingly, the treatment with Z-VAD-FMK, an inhibitor of pan-caspase activity, ameliorates the DNA fragmentation in vitrified oocytes, but it does not restore the altered oocyte development [92]. In humans, vitrification of oocytes during the MII stage induces changes in chromosomal and mitotic spindle configuration. Interestingly, this alteration is not observed when oocytes are pre-incubated at 37 °C [98].

The limited information regarding the molecular changes after cryopreservation can be overcome through new sequencing techniques since they provide a large amount of information related to gene expression, even at single-cell resolution. Recent evidence showed that oocytes submitted to either vitrification or slow-freezing procedures display transcriptomic alterations since both methods generate modifications in the gene expression profile of human MII oocytes; these changes are characterized by the downregulation of several genes related to embryo development, energetic pathways, DNA integrity, and cell cycle. Interestingly, these changes are more conspicuous in oocytes submitted to slow-freezing procedures [99,100]. However, there is evidence that vitrification can affect the imprinting of essential genes such as *GTL2* and *PEG3* [101] and decreases the expression of *PTEN*. In agreement, recent data of single-cell RNA sequencing demonstrated that the vitrification procedure of human oocytes impaired their gene expression when compared with fresh oocytes. On the other hand, distinct periods of cryopreservation did not alter gene expression of oocytes submitted to vitrification [102]. Although these findings have contributed to understanding the impact of cryopreservation methods on germ cells, a deeper analysis of the transcriptomic changes at single-cell resolution of embryos and the effect of slow-freezing procedures on embryos and gametes is still needed.

## 5. Molecular Alterations in Cryopreserved Sperm

Sperm cryopreservation induces physical changes that alter proteins that participate in several cellular processes related to membrane permeability, motility, metabolism, apoptosis, capacitation, and fertilization [103].

### 5.1. Cellular and Proteomic Changes in Sperm

Oxidative stress in human sperm affects lipid composition, proteins, and DNA, leading to reduced viability, motility, and fertility potential [104,105,106]; thus, oxidative stress has been proposed as an important damage mechanism during cryopreservation. Sperm freezing induces ROS production and decreases the levels of antioxidant factors, which correlate with lipid peroxidation, DNA fragmentation, apoptosis, and membrane damage after thawing [61,107,108]. For instance, in cryopreserved human sperm, the antioxidant enzyme SOD1 is decreased; this effect is also observed in buffalo and chicken sperm. Moreover, CPA modify proteins involved in metabolism and oxidative reactions even in unfrozen human sperm [103]. Thus, antioxidant treatment has emerged as a possible strategy to overcome these impairments on human sperm [61].

Antioxidant factors are proposed to counteract ROS production in cryopreservation. In semen of normozoospermic individuals, myoinositol supplementation during cryopreservation significantly increased the antioxidant capacity, although this treatment did not affect ROS levels [109]. Additionally, in mice sperm, L-carnitine supplementation decreased the ROS and protein carbonylation levels produced after thawing [110]. Further analysis showed that antioxidants ameliorate the motility, viability, and DNA integrity impairments of sperms that could be reflected in increased fertility rates [111,112]. Therefore, more studies regarding the potential benefits of antioxidant treatments should be performed.

ROS production and DNA damage in freeze-thawing processes lead to apoptosis and reduced sperm viability. Cryopreservation of human spermatozoa correlates with the activation of Caspases 3, 8, and 9, and mitochondrial membrane potential impairment, which induces the release of mitochondria-associated proteins [113]. Additionally, reduced protein levels of AIFM1 and cytochrome CYC2, and increases of Clusterin and Importin-1β, which are proteins involved in the mitochondrial apoptosis and DNA damage, have been described [103].

Cryopreservation is frequently associated with reduced sperm motility and fertilizing potential due to membrane, cytoskeletal, and acrosome impairments [114]. Increased protein levels of Annexins 1, 3, and 4 were observed in freeze–thawing sperm samples, suggesting the presence of cell membrane impairment. In addition, Tubulin-α 1A chain, which participates in spermatozoa motility, displays increased levels in cryopreserved samples [103]. Moreover, in another study, α-Tubulin was significantly increased in cryopreserved sperm samples with a marked increase at long storage periods [115].

Further studies provided consistent information related to the cytoskeletal defects and sperm motility after sperm cryopreservation. In a proteomic assay of cryopreserved human sperm, there was a decrease in the expression of Vimentin, Tektin-1, and Aconitate hydratase mitochondrial 2 (ACO2). Vimentin participates in the assembly and stabilization of cell surface domains in the spermatozoa, which are associated with fertilization; in this manner, low Vimentin expression may lead to acrosomal impairment in cryopreserved spermatozoa. Tektin-1 comprises a structural component of microtubules in cilia and flagella, carrying out a key role in sperm motility. ACO2 catalyzes the isomerization of citrate to isocitrate in the tricarboxylic acid cycle, which is crucial for ATP production; isocitrate content was significantly decreased in cryopreserved sperm; therefore, ATP-dependent sperm motility could be compromised [116]. In addition, reduced motility in cryopreserved sperm of healthy donors correlates with decreased levels of P34H. This protein plays a key role in the binding to the zona pellucida of the egg. Interestingly, lack of P34H in sperm membrane has been related to some cases of human infertility [115].

Recently, it was reported that slow-frozen human sperm show decreases in mRNA and protein levels of CATSPER2 and TEKT2, which are involved in sperm motility. CATSPER2 is a calcium ion channel found in spermatozoa flagellum, and its low expression has been found in subfertile men sperm with reduced motility. On the other hand, TEKT2 participates in microtubules formation, playing a critical role in flagella formation and development; mutations in the TEKT2 gene affect spermatozoa motility and lead to infertility [117]. On the other hand, Sperm-associated antigens SPAG5, SPAG7, and SPAG12 are genes related to the successful outcome in assisted reproduction and may correlate with in vitro fertilization failure since they are significantly decreased in cryopreserved human sperm [118].

### 5.2. Epigenetic Changes in Sperm

In addition, DNA fragmentation induced by freeze-thawing processes correlates with an increase of SUMOylation of Topoisomerase IIα, which is involved in DNA cleavage and repair [119]. Furthermore, human sperm samples exposed to DNA damage by cryopreservation, UV irradiation or H_2_O_2_, display DNA lesions in genes with distinct compaction patterns such as *HOXA3*, *HOXB5*, *SOX2*, and *β-GLOBIN*, even when prominent DNA fragmentation was not found. These results suggest that DNA damage is unrelated to the DNA compaction pattern modulated by sperm nuclear basic proteins [120]. However, Histone 4 levels increase after cryopreservation and thawing, suggesting the presence of alterations associated with chromatin remodeling and compaction [103]. Notably, cryopreserved sperm samples display DNA damage in the genes *PRM1*, *BIK*, *FSHB*, *PEG1/MEST*, *ADD1*, *ARNT*, *UBE3A*, and *SNORD116/PWSAS*, which are involved in fertilization, embryo development, and epigenetic syndromes [121].

These results provide consistent information stating the association of molecular changes with sperm cryopreservation, such as motility and impairments that compromise the fertilization in assistant reproduction techniques. More studies shall be performed to determine the real impact of cryopreservation procedures on sperm viability and functionality.

### 5.3. Genetic and Transcriptomic Changes in Sperm

DNA damage is extensively described in cryopreservation and thawing processes. Increased DNA fragmentation has been observed in cryopreserved sperm of both fertile and subfertile men [112,114,122,123]. A study reported that cryopreservation of sperm samples by vitrification displays less DNA fragmentation and impaired acrosomes than samples cryopreserved by slow-freezing procedures, suggesting that cryopreservation procedures differentially affect sperm samples [124]. Moreover, mitochondrial DNA fragmentation, assessed by Caspase 3 staining, increased in cryopreserved samples of fertile men [123]; however, in another study, no change in Caspase 3 activation was observed in both normozoospermic and nonnormozoospermic samples [114]. Interestingly, DNA fragmentation levels were comparable between samples with three freezing-thawing cycles and a single cycle when samples were refrozen in their original cryoprotectant medium [122].

In a transcriptomic assay of cryopreserved human samples, the reduced mRNA levels of *PRM1*, *PRM2*, and *PEG1/MEST* were considered sperm quality markers and *ADD1* as a pregnancy success marker [121].

Although it has been reported that human spermatozoa are transcriptionally silent, findings in other species suggest that transcriptional activity is possible in different contexts [125,126]. Transcriptomic alterations also have been reported in cryopreserved sperm from other mammals. *Bax* upregulation and *Bcl2* downregulation were related to enhanced apoptosis in cryopreserved mice sperm [110]. In boar, cryopreservation increased mRNA levels of *FTO* and decreased *METTL3*, *METTL14*, *ALKBH5*, and *YTHDF2*, which are transcripts involved in the posttranscriptional mRNA modification mediated by N6-methyladenosine [127]. In frozen–thawed bull sperm, the *RPL31* gene was upregulated [128].

MicroRNAs (miRNAs) repress mRNA translation, differential miRNAs expression was found in boar sperm, miR-98 was significantly upregulated in frozen samples, and the expression levels of its mRNA targets, *FAS* and *BCL2*, were significantly reduced; conversely, miR-22 was decreased in cryopreserved samples when compared with fresh sperm and its target, *PTEN*, presented the opposite pattern [129].

## 6. Perspectives

Nowadays, different methods are studied to improve cryopreservation outcomes by modifying CPA, freezing rate, and warming, trying to avoid the abrupt changes at the molecular level and thus improve viability [45].

During the last years, methods have been developed to optimize different protocols. For instance, vitrification increases apoptosis in embryonic stem cells, and Rho-associated coiled-coil kinase (ROCK) is involved in this process. Postvitrification treatment by inhibiting ROCK improves the survival of vitrified/thawed bovine oocytes, human embryonic stem cells (hESC), and bovine blastocysts [130,131,132]. ROCK inhibitors are essentials in the thawing of hESC since they decrease apoptosis, although they induce changes in their metabolism [133].

Many factors affect the survival of cryopreserved gametes and embryos, and its mechanisms remain unclear. Recent studies that applied a high hydrostatic pressure during pretreatment of oocytes, embryos, and murine blastocyst showed increased viability since it induces general adaptation and increases tolerance of various in vitro procedures [134], and may involve Heat shock proteins (HSP) production and mRNA stabilization [135,136].

Cryopreservation engages oxidative stress and cell injury, resulting in changes in oxidation of amino acids or nucleic acids, membrane peroxidation, apoptosis, and necrosis. Antioxidant treatment after thawing bovine embryos increases the chance of developing blastocyst, even of fine quality [137]. Indeed, α-tocopherol improves blastocyst yield from the postwarm bovine oocytes [138]. Some antioxidants affect genes involved in survival, apoptosis, and oxidative stress and increase the expression of anti-apoptotic genes such as *Bcl2l1* and *Bcl-2* and decrease pro-apoptotic genes such as *BAX/Bax* [139]. More research regarding the culture media used in cryopreservation techniques is necessary.

A new strategy is the freeze-all as an alternative to fresh embryo transfer during IVF cycles. It is based on the segmentation of ovarian stimulation, ovulation triggering, the elective cryopreservation of all viable embryos, and the transfer of vitrified-warmed embryos in subsequent natural or artificial cycles [140]. All cryopreserved embryos are used with a preimplantation genetic screening at the fifth day [141].

The impact of cryopreservation on human embryos and germ cells requires further investigation. Unfortunately, one of the major issues that halt the progress of this research area is the availability of mainly human embryos and oocytes. Cell reprogramming consists of converting somatic cells to induced pluripotent stem cells, giving rise to virtually any type of differentiated cell [142,143]. This technology represents a promising alternative to circumvent the lack of human samples since it could provide either sperm or oocytes derived from somatic cells for research. Recent studies have developed a protocol based on cell reprogramming to develop oogonia from somatic cells [144]; this advance could boost the survey on human germ cells, their cryopreservation and shows future directions toward this research area.

A recent article achieved ex utero culture of post-implantation mouse embryos from before gastrulation [145]. The cultured embryos recapitulate in utero development, and this technique could be used to further compare molecular changes during embryogenesis without the need for cryopreservation, at least in rodents.

Trained personnel carry out cryopreservation and involves multiple steps to process the biological material and toxic CPA. The whole process is manual and sometimes not reproducible. One solution could be the use of a mechanized approach with microfluidics and automation [146]. Microfluidic devices could decrease cell damage from ice crystals by gradually exchanging water and CPA during cooling and controlled rehydration during warming, reducing osmotic stress. Unfortunately, little evidence is available on the clinical advantages of automated vitrification to promote its routine use [147].

## 7. Conclusions

The molecular changes exerted by cryopreservation are a field barely explored. More information is required regarding activation of signaling pathways related to cell death, long-term effects on the offspring derived from cryopreserved embryos, sperm or oocytes, and proteomic, transcriptomic, and epigenetic alterations. The advent of new technologies such as single-cell RNA sequencing could provide valuable information to understand these modifications; thus, further application of cutting-edge techniques is required to unveil the molecular changes occurring in cryopreserved embryos and germ cells to identify potential molecular targets that could help to improve the cryopreservation procedures.

## Figures and Tables

**Figure 1 ijms-22-10864-f001:**
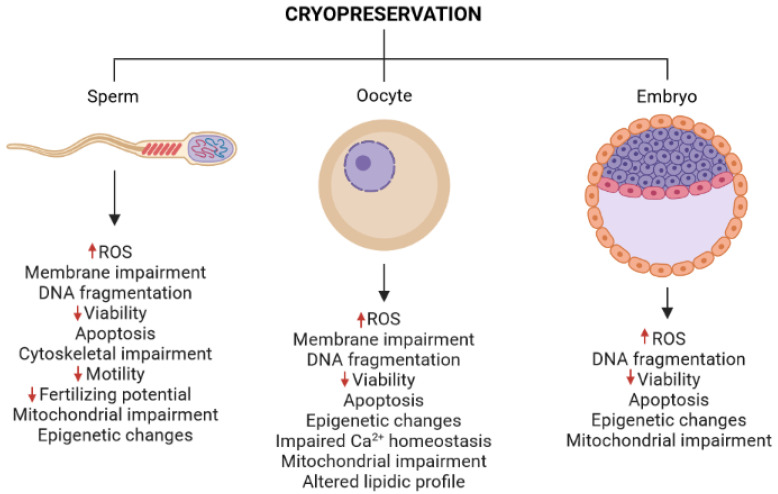
Effects of cryopreservation in germ cells and embryos. Cryopreservation of gametes and embryos induces molecular impairments related to metabolism, epigenetics, DNA integrity, cell death, and cytoarchitecture. ROS, reactive oxygen species. The red arrows represent increases (up arrows) or decreases (down arrows) in the indicated parameter. Figure elaborated with BioRender.

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
