# Peer review of "Cryopreservation of Gametes and Embryos and Their Molecular Changes"

_ijms, 2021, doi:10.3390/ijms221910864_

Round 1
Reviewer 1 Report
This is a short and simple review, certainly easy to read even for non-experts, but this is the problem, it does not add anything new to scientific speculation among experts. Given the scientific level of the journal, I don't think this review is suitable for publication
Author Response
Response to Reviewer 1 Comments
Point 1: This is a short and simple review, certainly easy to read even for non-experts, but this is the problem, it does not add anything new to scientific speculation among experts. Given the scientific level of the journal, I don't think this review is suitable for publication
Response 1: We are truly grateful for the generally positive comments on our review and regret that it was considered that it does not add anything new to the field. Unfortunately, the data regarding the molecular changes caused by cryopreservation of human embryos and germ cells are scarce. Therefore, we believe that this review will highlight the importance of this lack of information and contribute to identify key publications in order to design further experiments that could increase our knowledge of this topic. Trying to improve the quality of the manuscript, we modified it and added more information, including preclinical studies when there is little information about humans.
The Introduction now has a paragraph with the aim of this review.
In the Vitrification and Slow-Freezing Procedures chapter, the General Principles of Cryopreservation were added and abounded in the biological properties of cryopreservation and CPA.
Chapter 3 was divided into 3.1. Cellular Changes in Embryos, 3.2. Proteomic Changes, 3.3. Epigenetic Changes in Embryos, 3.4. Transcriptomic and Genomic Changes in Embryos, and 3.4. Offspring Changes in Embryos.
Chapter was 4 divided into 4.1. Cellular Changes in Oocytes, 4.2. Proteomic Changes in Oocytes, 4.3. Epigenetic Changes in Oocytes, and 4.4. Transcriptomic and Genomic Changes in Oocytes.
Chapter 5 was divided into 5.1. Cellular and Proteomic Changes in Sperm, 5.2. Epigenetic Changes in Sperm, and 5.3. Genetic and Transcriptomic Changes in Sperm.
In all these chapters, were now added changes at the mitochondrial level.
A Perspectives chapter was also added to discuss possible strategies to improve the viability of cryopreserved gametes and embryos without triggering molecular changes.
We hope that with these changes, the quality of the manuscript will be considered publishable.
Reviewer 2 Report
This review aims to provide a general framework on the molecular alterations of cryopreserved gametes and embryos, in my opinion this review is suitable for publication on IJMS but needs a major revision as many parts need to be reviewed and reorganized adequately, as it is difficult to follow.
INTRODUCTION
The introduction needs to be deeply revised and the aim of the review must be better stated.
Chapter 2 Vitrification and Slow-Freezing Procedures
This chapter should be organized in different sections (such as embryo, oocyte, and semen).
Add more information about the general principles of slow-freezing and cryopreservation.
Chapters 3, 4 and 5 are difficult to follow, it is necessary to treat the various molecular alterations (such as proteomic, transcriptomic, epigenetic, ecc….) individually.
CONCLUSIONS
The conclusions need to be revised considering the objective of the review and add future perspective of this fields (molecular changes) to improve the gamete and embryo cryopreservation.
Author Response
Point 1: This review aims to provide a general framework on the molecular alterations of cryopreserved gametes and embryos, in my opinion this review is suitable for publication on IJMS but needs a major revision as many parts need to be reviewed and reorganized adequately, as it is difficult to follow.
Response 1: We are truly grateful for the generally positive comments on our review. In order to improve the manuscript, we added information and reorganized some chapters.
Point 2: The introduction needs to be deeply revised and the aim of the review must be better stated.
Response 2: The Introduction was revised and we made some changes; the last paragraph now contains the aim of this review.
Point 3: Chapter 2 Vitrification and Slow-Freezing Procedures
This chapter should be organized in different sections (such as embryo, oocyte, and semen).
Add more information about the general principles of slow-freezing and cryopreservation.
Response 3: We added the section General Principles of Cryopreservation and abounded in the biological properties of cryopreservation and CPA. We then reorganized the section Effects of Cryopreservation in Germ Cells and Embryos that now provides general information, followed by a deeper analysis into the following chapters. To facilitate the comprehension of the new organization, we also changed the title of the figure legend that refers to the section mentioned above.
Point 3: Chapters 3, 4 and 5 are difficult to follow, it is necessary to treat the various molecular alterations (such as proteomic, transcriptomic, epigenetic, ecc….) individually.
These chapters were divided into the following sections.
Chapter 3: 3.1. Cellular Changes in Embryos, 3.2. Proteomic Changes, 3.3. Epigenetic Changes in Embryos, 3.4. Transcriptomic and Genomic Changes in Embryos, and 3.4. Offspring Changes in Embryos.
Chapter 4: 4.1. Cellular Changes in Oocytes, 4.2. Proteomic Changes in Oocytes, 4.3. Epigenetic Changes in Oocytes, and 4.4. Transcriptomic and Genomic Changes in Oocytes.
Chapter 5: 5.1. Cellular and Proteomic Changes in Sperm, 5.2. Epigenetic Changes in Sperm, and 5.3. Genetic and Transcriptomic Changes in Sperm.
All these chapters were now added changes at the mitochondrial level in the sections of cellular changes.
Point 4: The conclusions need to be revised considering the objective of the review and add future perspective of this fields (molecular changes) to improve the gamete and embryo cryopreservation.
The conclusions were revised and we added a Perspectives chapter to discuss possible strategies to improve the viability of cryopreserved gametes and embryos without triggering molecular changes.
We hope that with these changes, the quality of the manuscript will be considered publishable.
Reviewer 3 Report
I appreciate the authors for writing this review. It is an interesting review and almost covers all aspects of gametes and embryo cryopreservation. However, molecular changes on cryopreserved spermatozoa receive a bigger proportion compare to embryos and oocytes. I am aware that sperm has been extensively studied in all aspects including the impact of cryopreservation on its molecular changes.
I would like to see more information on both oocyte and embryos, but it may be limited by page number.
Author Response
Point 1: I appreciate the authors for writing this review. It is an interesting review and almost covers all aspects of gametes and embryo cryopreservation. However, molecular changes on cryopreserved spermatozoa receive a bigger proportion compare to embryos and oocytes. I am aware that sperm has been extensively studied in all aspects including the impact of cryopreservation on its molecular changes.
I would like to see more information on both oocyte and embryos, but it may be limited by page number.
Response 1: We are truly grateful for the generally positive comments on our review. Trying to improve the quality of the manuscript, we modify it and add more information.
Now, we divide the chapters into sections to make them easier to follow up and expand on oocytes and embryos.
Chapter 3 (Molecular Alterations in Cryopreserved Embryos) was divided into 3.1. Cellular Changes in Embryos, 3.2. Proteomic Changes, 3.3. Epigenetic Changes in Embryos, 3.4. Transcriptomic and Genomic Changes in Embryos, and 3.4. Offspring Changes in Embryos.
Chapter 4 (Molecular Alterations in Cryopreserved Oocytes) was divided into 4.1. Cellular Changes in Oocytes, 4.2. Proteomic Changes in Oocytes, 4.3. Epigenetic Changes in Oocytes, and 4.4. Transcriptomic and Genomic Changes in Oocytes.
Chapter 5 (Molecular Alterations in Cryopreserved Sperm) now is 5.1. Cellular and Proteomic Changes in Sperm, 5.2. Epigenetic Changes in Sperm, and 5.3. Genetic and Transcriptomic Changes in Sperm.
In all these chapters, changes at the mitochondrial level were added.
A Perspectives chapter was also added to discuss possible strategies to improve the viability of cryopreserved gametes and embryos without triggering molecular changes.
As the reviewer mentioned, there is more information on cryopreservation in sperm; however, we added information about preclinical studies when there was no information concerning human species. We hope that new changes compensate for this disproportion in the manuscript.
We hope that with these changes, the quality of the manuscript will be considered publishable.
Round 2
Reviewer 2 Report
The manuscript has been sufficiently improved and it is potentially publishable in present form, although there are some parts that are difficult to follow. In this regard I am not qaulified enough to judge if it is necessary extensive or moderate editing of english.